# Cancer Vaccination against Extracellular Vimentin Efficiently Adjuvanted with Montanide ISA 720/CpG

**DOI:** 10.3390/cancers14112593

**Published:** 2022-05-24

**Authors:** Karlijn van Loon, Elisabeth J. M. Huijbers, Jan David de Haan, Arjan W. Griffioen

**Affiliations:** Laboratory, Cancer Center Amsterdam, Department of Medical Oncology, Medical Center, VU University, Amsterdam UMC, 1081 HV Amsterdam, The Netherlands; k.vanloon@amsterdamumc.nl (K.v.L.); e.huijbers@amsterdamumc.nl (E.J.M.H.); j.d.dehaan@amsterdamumc.nl (J.D.d.H.)

**Keywords:** immunotherapy, vaccination, conjugate vaccine, adjuvant, angiogenesis, tumor vasculature, extracellular vimentin

## Abstract

**Simple Summary:**

Vaccination against specific proteins in the tumor vasculature has already shown promising results in several preclinical studies. However, the efficacy of vaccination highly depends on the adjuvant used. This study aimed to assess the potential use of the biodegradable adjuvant Montanide ISA 720 in combination with our vaccine against extracellular vimentin, a protein specifically secreted by the tumor vasculature. Compared to the potent but toxic Freund’s adjuvant, Montanide showed a comparable immune response and tumor growth inhibition in a preclinical vaccination experiment in mice, especially when supplemented with the immune stimulatory molecule CpG. We also observed that vaccination reduced the blood vessel count and increased the infiltration of immune cells. We conclude that Montanide ISA 720 shows potential to be used as an adjuvant for vaccination against extracellular vimentin for future clinical studies in cancer patients.

**Abstract:**

Extracellular vimentin is a specific marker of the tumor vasculature, where it is secreted by tumor endothelial cells. Vaccination with a conjugate vaccine targeting extracellular vimentin was previously shown to induce a potent humoral immune response and tumor growth inhibition in mice. These data were obtained by vaccination using the toxic Freund’s adjuvant (FA) and are therefore not directly translatable into the clinic. In the present study, we aimed to investigate the potential of the biodegradable Montanide ISA 720 adjuvant. We tested Montanide either alone (MN) or supplemented with CpG 1826 (MN-C). Both adjuvant compositions, as well as FA, resulted in a significant tumor growth inhibition and decreased vessel density in the B16F10 melanoma tumor model. Vaccination of mice with either FA or MN-C resulted in an equally potent humoral immune response towards vimentin, while the antibody titers obtained with MN alone were significantly lower compared to FA. Vaccination coincided with the infiltration of immune cells. The highest number of intratumoral immune cells was seen in tumors from the MN-C group. Therefore, we conclude that Montanide ISA 720 supplemented with CpG allows efficient vaccination against extracellular vimentin, which is a prerequisite for the transfer of the vaccine into the clinic.

## 1. Introduction

Over the past decade, immunotherapy has revolutionized cancer treatment, mainly by immune checkpoint inhibitors, such as nivolumab (anti-PD1) and ipilimumab (anti-CTLA4) [1], but also by adoptive immune cell therapies [2] and therapeutic cancer vaccines [3]. Inhibition of angiogenesis is widely introduced in the clinical management of cancer, and the combination of angiogenesis inhibitors with immunotherapy has recently come of age [4,5], increasing the interest in the development of new anti-vascular strategies.

Crucial in the development of angiogenesis inhibitors is the identification of specific and selective targets in the tumor vasculature [6,7,8,9]. We recently showed that vimentin is specifically excreted by angiogenic tumor endothelial cells and plays a stimulatory role in the process of tumor angiogenesis, making it a promising candidate for anti-cancer vaccination [9]. Vaccines targeting tumor endothelial-specific antigens need to be able to break immune tolerance against the self-antigen. To achieve this, conjugate protein vaccine technology can be used. This approach makes use of conjugation of the target of interest to a foreign protein, such as the truncated form of bacterial thioredoxin (TRXtr) [10]. The conjugate vaccine technology has been shown to efficiently induce an antibody-based immune response against vascular-specific self-proteins [7,8,10,11,12]. It is important to mention that in addition to an efficient immunization strategy, the co-administration of a potent adjuvant is needed to generate this immune response. Freund’s adjuvant (FA) is currently the gold standard in preclinical vaccination studies. This adjuvant is known to be highly immunostimulatory, but unfortunately, it also generates serious side effects and is therefore not approved for clinical use. Montanide-based adjuvants are currently being investigated in clinical studies.

In the present study, we investigated the potential of the squalene-based Montanide ISA 720 adjuvant in combination with the conjugate vaccine targeting extracellular vimentin, referred to as TRXtr-Vim. Montanide ISA 720 has previously been shown to drive the generation of antibodies against the extra domain A (ED-A) and B (ED-B) of fibronectin when combined with single-stranded CpG oligodeoxynucleotide 1826, an agonist of the murine toll-like receptor (TLR)-9 [8,13]. Similar to FA, the use of Montanide generates a water-in-oil vaccine emulsion, resulting in a slow-release depot of antigens at the injection site [14]. In this study, we tested Montanide either alone (MN) or supplemented with CpG 1826 (MN-C). Both tested Montanide compositions resulted in the induction of anti-vimentin antibodies and significant tumor growth reduction in the murine B16F10 melanoma model, as compared to mice that received a control vaccine. MN-C resulted in humoral immunity comparable to FA, which was characterized by a high abundance of IgG2b and IgG2c antibodies. Therefore, we conclude that Montanide ISA 720 supplemented with CpG can replace the toxic Freund’s adjuvant in a conjugate vaccine targeting extracellular vimentin. Since the adjuvant Montanide ISA 720 alone or supplemented with CpG is already used in several clinical trials in humans, the results of this study facilitate further evaluation of the TRXtr-Vim vaccine in the clinic.

## 2. Materials and Methods

### 2.1. Vectors and Vaccine Production

The recombinant thioredoxin (TRX, 13 kDa), vimentin (Vim, 55 kDa), and TRXtr-Vim (61 kDa) proteins were produced and purified as previously described [9]. The protein-coding sequences of all three proteins were cloned into the multiple cloning site of the pET21a(+) (Novagen, Merck Millipore, Darmstadt, Germany) expression vector (Appendix A–C). Reference sequences were retrieved from UniProt: POAA25-1 for TRX and P20152-1 for Vim. The pET21-TRX plasmid was transformed into *E. coli* strain Rosetta Gami (DE3) (Novagen, Merck Millipore), while the pET21-Vim and pET21-TRXtr-Vim vectors were transformed into *E. coli* strain BL21 (DE3) (Novagen, Merck Millipore) and stored as glycerol stocks at −80 °C.

LB medium containing 100 µg/mL ampicillin (Amp, Sigma Aldrich, Zwijnsdrecht, The Netherlands, Cat. A9518-5G) was inoculated with bacterial stocks and grown overnight at 37 °C at 200 rpm. The next day, cultures were diluted 1:3 with LB medium containing Amp, and protein expression was induced with 1 mM isopropyl β-D-1-thiogalactopyranoside (IPTG, Serva, Heidelberg, Germany, Cat. 26600.04) at 37 °C for 4 h. Cultures were centrifuged for 10 min at 4500 rpm, and pellets from 50 mL culture were resuspended in 5 mL sonication buffer; induced pellets containing TRX protein were resuspended in 5 M urea (Across Organics, Landsmeer, The Netherlands, Cat. 140750050), while Vim- and TRXtr-Vim containing pellets were resuspended in sonication buffer containing 2 M urea in PBS, 20% glycerol (VWR Chemicals, Amsterdam, The Netherlands, Cat. 24386.398), 0.1 µM EDTA (J.T.Baker, Deventer, The Netherlands, Cat. 1073), and 1% Triton X-100 (Amresco, Solon, OH, USA, Cat. M143). Bacterial suspensions were sonicated for 15 cycles (20 s on and 30 s off) with an amplitude of 22–24 microns using the Soniprep (150 MSE, London, UK). For Vim and TRXtr-Vim, 1 mM phenylmethylsulphonyl fluoride (PMSF, Sigma-Aldrich, Saint Louis, MO, USA, Cat. 93482) was added to the supernatant after sonication to prevent protein degradation. Finally, tubes were centrifuged at 4500rpm for 10 min, and the supernatant was stored at −20 °C until protein purification.

All three proteins contain a C-terminal His-tag (Appendix A–D), enabling purification using Ni-Affinity chromatography. Per 10 mL of thawed supernatants containing Vim or TRXtr-Vim, 300 µL Ni-NTA agarose slurry (Qiagen, Venlo, The Netherlands, Cat. 1018244) was added and incubated at 4 °C overnight. After centrifugation, the agarose beads were washed with wash buffer (PBS pH 7.0, 1M NaCl, 0.05% Tween-20) and transferred to a column with a glass filter (Sartorius Stedim Biotech, Göttingen, Germany, Cat. 82121-011-03). The column was washed with four 150µL fractions of 200 mM imidazole (J.T.Baker, Deventer, The Netherlands, Cat. 1747) and dissolved in 20 mM Tris pH 8.0 with 0.1 M NaCl, and the proteins were eluted with four 150 µL fractions of elution buffer (100 mM NaH_2_PO_4_ (Merck, Darmstadt, Germany, Cat. 1.06346.1000), 10 mM Tris·Cl, 8M urea, pH 4.5) (Appendix A). To purify the TRX protein, 10 mL of supernatant was incubated with 300 µL Ni-NTA agarose slurry in the presence of 10 mM imidazole at 4 °C overnight. The next day, the agarose beads were washed with 0.1 M NaCl, 0.1% Tween-20 in PBS and transferred to a column with a glass filter. The column was washed with 10 fractions of 20 mM imidazole in 20 mM Tris pH 8.0 with 0.1 M NaCl, and protein was eluted with four 150 µL fractions of 200 mM imidazole (Appendix A).

Protein content in elution fractions was visualized by SDS-PAGE using Mini-PROTEAN^®^ TGX 4–20% precast polyacrylamide gels (Bio-Rad, Veenendaal, The Netherlands, Cat. 4561094), stained with colloidal Coomassie brilliant blue G250 (Polysciences, Hirschberg an der Bergstrasse, Germany, Cat. 03707-50) (Appendix A). Fractions containing the protein of interest were dialyzed using 3.5 K SnakeSkin (Thermo Fisher Scientific, Landsmeer, The Netherlands, Cat. 68035) for TRX and 10K SnakeSkin (Thermo Fisher Scientific, Cat. 88245) for Vim and TRXtr-Vim. TRX protein fractions were directly dialyzed in PBS at 4 °C overnight. Vim and TRXtr-Vim proteins were dialyzed against 4 M urea in PBS at 4 °C overnight, followed by stepwise dialysis in 3 M (2 h), 2.5 M (2 h), and 2 M (2 h) urea. Final protein concentrations were determined with a Micro BCA Protein Assay (Thermo Fisher Scientific, Cat. 23235).

### 2.2. Animal Studies

All animal experiments were performed in accordance with Dutch guidelines and law on animal experimentation and were approved by the Centrale Commissie Dierproeven (CCD), registration no. DEC AngL13-02 and CCD AVD114002016576. Work protocols were approved by the VU-VUmc animal welfare body.

In the first in vivo experiment, three different experimental groups were compared, consisting of five 8-week-old female C57BL/6JOlaHsd mice (Envigo) each. All mice were vaccinated three times subcutaneously in the groin with an interval period of two weeks between immunizations. The first group was immunized with 30µg vimentin protein only and the second group with 40 µg of the conjugate vaccine TRXtr-Vim only. The final group was immunized with an emulsion containing 20 µg of TRXtr-Vim, mixed with Freund’s complete adjuvant (FCA, Sigma Aldrich, Zwijnsdrecht, The Netherlands, Cat. F-5881) in a 1:1 ratio. For the second and third vaccination, FCA was replaced by Freund’s incomplete adjuvant (FIA, Sigma Aldrich, Cat. F-5506). Blood samples were taken from the tail vein before the first vaccination and 1 week after each vaccination. Blood samples were coagulated overnight at 4 °C and centrifuged at 7000 rpm for 10 min at 4 °C in a microcentrifuge. The supernatant (serum) was removed, centrifuged again to remove additional debris, and stored at −20 °C until use.

Subsequently, the in vivo adjuvant comparison study was performed in duplicate and is indicated as ‘Study I’ and ‘Study II’ in other sections. Each treatment/control group contained ten 8-week-old female C57BL/6JOlaHsd mice (Envigo). Mice were immunized four times subcutaneously in the left groin with an interval period of two weeks between immunizations. Each vaccine emulsion contained 75 µg TRXtr-Vim or 10 µg TRX protein in a volume of 50 µL PBS (TRX) or 2 M urea (TRXtr-Vim), mixed with 50 µL adjuvant (ratio 1:1, aqueous phase: oil phase); total injection volume was 100 µL. In total, three different adjuvant compositions were compared. The first group received a primer vaccination with an emulsion containing FCA, followed by booster vaccinations with FIA. One study group received MN720 supplemented with 50 µg phosphorothioate CpG oligonucleotide 1826 (Eurogentec, Seraing, Belgium) per mouse during all 4 vaccinations, and the final group received MN720 alone, without any supplementation. All emulsions were mixed for 30 min on a Vortex-Genie 2 (Thermo Fisher Scientific) at full speed. Blood samples were taken from the tail vein at the start of the experiment before the first vaccination, one week after each vaccination, and at the end of the experiment. Blood samples were coagulated overnight at 4 °C, centrifuged, and the serum was stored at −20 °C until use.

Murine B16F10 melanoma tumor cells (ATCC CRL-6475) were maintained in DMEM (Gibco, Waltham, MA, USA, Cat. 41965-039) supplemented with 1% of antibiotics (Penicillin/Streptomycin, Life Technologies, Carlsbad, CA, USA, Cat. 15140-122), 10% newborn calf serum (NBCS, Biowest, Nuaillé, France, Cat. S0750), and 2 mM L-glutamine (Roth, Karlsruhe, Germany Cat. HN08.2). One week after the last immunization, 1 × 10^5^ B16F10 melanoma cells were inoculated subcutaneously in the left flank of C57BL/6 mice in a total volume of 100 μL (10% culture medium/PBS). Tumor growth was measured by calipers, and tumor volume was calculated with the formula: width^2^ × length × π/6. In Study I, two mice in the MN-C group and one mouse in the MN group showed no tumor take and were excluded from tumor growth data. Similarly, in Study II, one mouse in the MN-C group was excluded. In addition, two TRX control mice were sacrificed before tumor cell injection due to health issues. At the end of the experiment, mice were euthanized, and tumors and organs were removed and stored in 4% PFA/PBS (Electron Microscopy Sciences, Hatfield, PA, USA, Cat. 15710) overnight and consecutively paraffin-embedded.

### 2.3. Immunohistochemistry

Paraffin-embedded tumor tissues obtained during in vivo Study II were sectioned (3–5 µm) with a microtome (Leica, Nieuw-Vennep, The Netherlands). Sections were dried at least 24 h at 37 °C. Tumor necrosis was analyzed using hematoxylin and eosin (HE) staining. For further analyses regarding vessel density and immune cell content of the B16F10 tumor, tissue sections were used for immunohistochemistry. Before deparaffinization, tissue sections were incubated at 60 °C for 1 h, followed by 10 min at 56 °C on a heating plate. Slides were deparaffinized with xylene and consecutively rehydrated in 100%, 96%, and 70% ethanol and PBS. Endogenous peroxidase activity was blocked with a 15 min incubation in 0.3–3% hydrogen peroxidase in PBS (VWR Chemicals) at RT. Antigen retrieval was performed in citrate buffer (10 mM, pH 6.0) by autoclaving. For CD3 staining, TRIS/EDTA pH 9.0 buffer was used during antigen retrieval. Blocking was performed using 3% bovine serum albumin (BSA, Roche, Penzberg, Germany, Cat. 10735086001) for CD31, CD45, and CD11b or 20% horse serum (Sigma-Aldrich, Saint Louis, MO, USA, Cat. H1138) in PBS for vimentin for 1h at RT. A blocking solution of 4% BSA with 5% normal goat serum (Sigma-Aldrich, Saint Louis, MO, USA, Cat. G6767) was used for blocking in the CD3 staining procedure. For anti-vimentin staining on mouse sections, additional blocking with Fab fragments (Abcam, Cambridge, UK, Cat. ab6668) in a concentration of 50 µg/mL was performed for 2 h at RT.

Next, slides were incubated with the following primary antibodies: anti-CD31 rat anti-mouse IgG2a (1:50, Dianova, Hamburg, Germany, Cat. DIA-310M, clone SZ31), anti-CD45 rabbit anti-mouse (1:100, Abcam, Cat. Ab10558), anti-CD11b rabbit anti-mouse (1:4000, Abcam, Cat. Ab13357), anti-CD3 rabbit anti-mouse (1:600, Neomarkers, Fremond, CA, USA, Cat. RM-9717-S), and mouse anti-vimentin (Santa Cruz Biotechnology, Dallas, TX, USA, Cat. sc-373717, Clone E5, 1:50 dilution). After overnight incubation with primary antibodies at 4 °C, tissue sections were washed in PBS and incubated with biotinylated secondary antibodies for 45 min at RT. The following biotinylated secondary antibodies were used: swine anti-rabbit (1:500, Dako, Glostrup, Denmark, Cat. E0353), donkey anti-rat (1:500, Jackson lab, Baltimore, PA, USA, Cat. AB2340650), and goat anti-mouse (1:500, Dako, Cat. E0433). Slides were again washed in PBS and incubated with Strep-HRP (Dako, Cat. P0397) in 0.5% BSA for 30 min at RT. For CD3 staining, a goat anti-rabbit HRP polymer (Abcam, Cambridge, UK, Cat. Ab214880) was used. After washing with PBS, 3,3′-diaminobenzidine tetrahydrochloride hydrate (DAB, Sigma-Aldrich, Cat. D5637) staining was performed for up to 8 min at RT. Sections were washed with PBS and counterstained with Mayer’s hematoxylin (VWR Chemicals, Amsterdam, The Netherlands, Cat. 10047105) (1:4 diluted in 5 mM citrate buffer pH6.0) for 30 s. The reaction was stopped under running tap water for 10 min, and sections were dehydrated consecutively with 70%, 96%, and 100% ethanol. Finally, slides were incubated in xylene and mounted with Quick D mounting medium (Klinikpath, Duiven, The Netherlands, Cat. 7280) and covered with a coverslip. Sections were visualized with a 200× magnification on an Olympus BX50 microscope. Pictures were taken with a CMEX DC 5000C camera. The number of blood vessels or immune cells was counted per tissue core, with three fields in total per section. One tumor from the MN-C group and one tumor from the MN group were extensively necrotic and were excluded for CD31, CD45, CD11b, and CD3 quantification.

To validate vimentin expression in the tumor vasculature of human tumors, a tissue microarray was generated in collaboration with the biobank at the pathology department in Amsterdam UMC, location VUMC. The microarray contains tumors of stage 1 and stage 3, according to TNM classification. Tissue cores with a 1mm diameter were obtained from formalin-fixed and paraffin-embedded (FFPE) blocks using the TMA Grandmaster. The TMA tissue block was sectioned and stained using an anti-vimentin antibody (Santa Cruz Biotechnology, Dallas, TX, USA, Cat. sc-373717, clone E5, 1:50 dilution), as described above. No Fab fragments were needed during the staining procedure.

### 2.4. ELISA

Indirect ELISA was performed as described previously [9]. Volumes per well were 50 µL, unless indicated otherwise. The 96-well ELISA plates (Thermo Fisher Scientific, Landsmeer, The Netherlands, Cat. 442404) were coated with 4 μg/mL recombinant vimentin (Vim) protein (see section protein production and purification) in 0.5 M urea in PBS for 1 h at 37 °C. To determine the antibody titers against TRX, ELISA plates were coated with TRX-gal1 according to Saupe et al. [15]. Next, wells were blocked with 4% non-fat dry milk (Bio-Rad, Veenendaal, The Netherlands, Cat. 1706404) in PBS for 1h at 37 °C using 100 µL per well. Plates were washed one time with PBS for 1 min to remove the excess blocking solution.

Mouse sera were briefly centrifuged and diluted 1:10 in 4% non-fat dry milk in PBS. For TRX antibody titers, 5 mouse sera were pooled per measurement to reduce the amount of serum needed. Sera were further diluted in Rosetta Gami extract, which is used to reduce the non-specific binding of serum antibodies to the recombinantly produced vimentin coating. Rosetta Gami extract was prepared by overnight culturing of Rosetta Gami DE3 pET21a-TRX bacteria at 37 °C at 200 rpm. The next day, the culture was centrifuged for 20 min at 4500 rpm, and the culture supernatant was discarded. Bacterial pellets were resuspended in PBS and sonicated for 15 cycles (20 s on and 30 s off) with an amplitude of 22–24 microns. After sonication, tubes were centrifuged at 4500 rpm for 10 min, and the supernatant was stored at −20 °C to be used as Rosetta Gami extract during ELISA.

Diluted sera were incubated in the 96-well plate for 45 min at 37 °C, followed by four washing steps with PBS. After washing, plates were incubated with biotinylated polyclonal goat anti-mouse Ig (Dako Cytomation, Glostrup, Denmark, Cat. E0433) and diluted 1:2000 in 0.01% PBS-T for 45 min at 37 °C, followed by four washing steps with PBS. Thereafter, wells were incubated with streptavidin–horseradish peroxidase (Dako Cytomation, Glostrup, Denmark, Cat. P0397) for 30 min at 37 °C, which was also diluted 1:2000 in 0.01% PBS-T. After four washes with PBS, wells were incubated with TMB substrate (Sigma-Aldrich, Saint Louis, MO, USA, Cat. T-8665) for 5 min, and absorbance was measured at 655 nm using a Biotek Synergy HT microplate reader (Biotek, Bad Friedrichshall, Germany). Antibody titers were determined by calculating at which serum dilution the titration curve would give an OD value of 0.2. This threshold was set based on the background signal observed in blank control wells and wells incubated with serum retrieved from mice vaccinated with the control TRX vaccine.

To determine the antibody isotypes of the anti-vimentin antibodies in mouse sera, a similar ELISA protocol was used as described above. However, for each measurement, sera of five different mice were pooled, and blocking was performed using 3% bovine serum albumin (BSA, Roche, Penzberg, Germany, Cat. 10735086001). In addition, the following isotype-specific biotinylated goat anti-mouse secondary antibodies were used: IgG1 (Southern Biotech, Birmingham, AL, USA, Cat. 1070 08), IgG2b (Southern Biotech, Birmingham, AL, USA, Cat. 1090 08), IgG2c (Southern Biotech, Birmingham, AL, USA, Cat, 1079 08), IgG3 (Southern Biotech, Birmingham, AL, USA, Cat. 1100 08), IgM (Southern Biotech, Birmingham, AL, USA, Cat. 1020 08), and IgA (Southern Biotech, Birmingham, AL, USA, Cat. 1040 08). Listed antibodies were diluted 1:2500 in 0.01% PBS-T and incubated for 45 min at 37 °C.

### 2.5. Surface Plasmon Resonance Using Biacore

All the surface plasmon resonance (SPR) experiments were done using Biacore T200 (Cytiva). Recombinant murine Vim protein was immobilized on the CM5 chip (Cytiva, Freiburg im Breisgau, Germany) surface using amine coupling at pH 4.0. All the serum samples obtained from Study I were injected over the measurement and reference cells at a flow rate of 20 μL/min for 120 sec in 2 mM Na Phosphate buffer pH 7.5, 0.05% Tween-20. To normalize the anti-vimentin antibody input from all sera, the serum dilution was normalized based on the determined anti-vimentin antibody titer by ELISA. At the end of each measurement cycle, the surface was regenerated by subsequent injections of 10 mM Glycine pH 1.5 and 0.5 M urea. Injection of a commercial anti-mouse vimentin antibody (Clone E5, Santa Cruz Biotechnology, Dallas, TX, USA, Cat. sc-373717) was used as the positive control. All samples were run in quadruplicate, and for each group, one mouse was excluded due to high variation between the technical replicates. Initial data analysis was done using Biacore T200 Evaluation Software v 3.2 (Cytiva, Freiburg im Breisgau, Germany).

### 2.6. LEGENDplex

The LEGENDplex™ MU Th Cytokine Panel (12-plex) w/ VbP V03 (BioLegend, London, United Kingdom, Cat. 741044) was used for quantification of cytokines in mouse sera, collected one week after the fourth vaccination from both Studies I and II. For FA, three out of 20 mice did not have a sufficient amount of serum for analysis, as well as one mouse from the MN-C-vaccinated group. Measurements were performed according to the manufacturer’s protocol and were read on a FACSCalibur (BD Biosciences, Franklin Lakes, NJ, USA). Data were analyzed using LEGENDplex™ Data Analysis Software Suite.

### 2.7. Splenocyte Isolation, Restimulation, and FACS Analysis

Spleens from mice immunized with TRX or TRXtr-Vim in combination with Freund’s adjuvant were isolated and cut into small pieces and mechanically dissociated. Splenocytes were passed through a 70 μm cell strainer (Corning, Amsterdam, The Netherlands, Cat. 431751) and centrifuged for 5 min at 1500 rpm. Red blood cells were lysed using a buffer containing 150 mM NH_4_Cl, 10 mM KHCO_3_, and 100 mM EDTA (pH = 7.4) for 3 min. Splenocytes were washed and resuspended in RPMI-1640 (Biowest, Nuaillé, France, Cat. L0495-500) medium supplemented with 1% of antibiotics (Penicillin/Streptomycin, Life Technologies, Carlsbad, CA, USA, Cat. 15140-122), 10% newborn calf serum (NBCS, Biowest, Cat. S0750), 2 mM L-glutamine (Roth, Karlsruhe, Germany, Cat. HN08.2), and 50 μM 2-mercaptoethanol (Gibco, Waltham, MA, USA, Cat. 31350-010). Isolated splenocytes were seeded in U-bottom 96-well plates with 2×10^6^ cells/well and were restimulated ex vivo for 48 h with or without recombinant endotoxin free vimentin (see section vectors and vaccine production) at a concentration of 20 µg/mL. Endotoxin was removed while the vimentin protein was bound to the Ni-agarose by washing the column with 20–50 column volumes of PBS-0.1% Triton X-114 (Sigma-Aldrich, Zwijnsdrecht, The Netherlands, Cat. X114-100 mL) at 4 °C in the cold room. Thereafter, the column was washed with 5–20 column volumes of PBS at 4 °C to remove the Triton X-114. Endotoxin levels were determined by LAL-assay (Thermo Scientific, Landsmeer, The Netherlands, Pierce Chromogenic Endotoxin Quant kit, Cat. A39552S). During the final 6 h of stimulation, Brefeldin A (BioLegend, London, United Kingdom, Cat. 420601) was added to all wells. After restimulation, cells were transferred to a V-bottom 96-well plate and washed with PBS. Cells were resuspended in Fc blocking solution (BioLegend, London, United Kingdom, Cat. 101320) for 10 min at room temperature. Next, cells were incubated with anti-mouse CD4-AF700 (1:200, BioLegend, London, United Kingdom, Cat. 100429), anti-mouse CD8b-PE (1:200, BioLegend, Cat. 126627), and Zombie Aqua fixable viability dye (1:200, BioLegend, London, United Kingdom, Cat. 423101) diluted in PBS for 20 min on ice. Cells were washed with PBS and fixed with 4% paraformaldehyde (PFA, Electron Microscopy Sciences, Cat. 15710) for 15 min on ice. After washing, cells were permeabilized using an intracellular staining permeabilization wash buffer (BioLegend, London, United Kingdom, Cat. 421002). Cell suspensions were then incubated with anti-mouse IFNγ-APC (1:200, BioLegend, London, United Kingdom, Cat. 505809) and TNFα-PE (1:200, BioLegend, London, United Kingdom, Cat. 506305) diluted in permeabilization buffer for 30 min at room temperature. Cells were washed twice with the permeabilization buffer and finally resuspended in 100 μL PBS. Samples were measured using the LSRII Fortessa (BD Biosciences, Franklin Lakes, NJ, USA) flow cytometer. Data analysis was performed with the FlowJo V10.8.1 software (Tree Star).

### 2.8. Statistics

All statistical tests were executed using GraphPad Prism 9.1.0 (GraphPad Software Inc., La Jolla, CA, USA), and * *p* < 0.05, ** *p* < 0.01, *** *p* < 0.001, and **** *p* < 0.0001 were considered statistically significant. Error bars represent the standard error of the mean (SEM) unless specified otherwise. For comparison of tumor growth curves and antibody isotype distributions, a two-way ANOVA was used. Other comparisons for different treatment groups were tested using a Kruskal–Wallis test with Dunn’s multiple comparison testing. Correlation plots were tested for statistical significance using simple linear regression analysis.

## 3. Results

### 3.1. Development of a Vaccine against Extracellular Vimentin and Preclinical Study Setup

Extracellular vimentin was previously shown to be a specific and promising target of the tumor vasculature [9]. To validate vimentin tissue expression, a tissue microarray (TMA) was generated. Immunohistochemistry confirmed a broad expression of vimentin in several types and stages of cancer, such as colorectal adenocarcinoma (Figure 1A). To generate a humoral immune response against the vascular target extracellular vimentin, a conjugate vaccine was generated consisting of truncated bacterial thioredoxin (TRXtr) and murine vimentin (Vim) (Figure 1B and Appendix A–D). After vaccine injection, the fusion protein is internalized by antigen-presenting cells and presented on MHC-II molecules, whereby TRXtr-specific T cells become activated. These TRXtr-specific T cells will in turn activate vimentin-specific B cells, which results in an antibody response against extracellular vimentin (Appendix A).

In an in vivo mouse experiment, we vaccinated mice with vimentin protein or TRXtr-Vim protein alone (without the addition of adjuvant) or TRXtr-Vim combined with Freund’s adjuvant (FA) (Figure 1C). Only mice vaccinated with TRXtr-Vim combined with FA showed a potent anti-vimentin antibody response, indicating the requirement for a potent adjuvant in our vaccination strategy.

We previously showed that the TRXtr-Vim vaccine, combined with FA, significantly inhibited tumor growth in the B16F10 melanoma model [9]. This murine B16F10 tumor model expresses vimentin in the tumor vasculature and surrounding matrix (Appendix A). To investigate whether the squalene-based adjuvant Montanide ISA 720 is equally potent as FA, we combined the TRXtr-Vim with either FA, Montanide adjuvant alone (MN), or Montanide with the addition of CpG 1826 (MN-C) (Figure 1D). CpG oligodeoxynucleotides activate TLR9, while Mycobacteria ligands, as found in Complete Freund’s adjuvant, can bind the extracellular receptors Mincle [16], Dectin-1 [17], and TLR2 [17]. In addition, it can activate the intracellular TLR4, TLR9, and TLR8 receptors [18]. Both CpG and Mycobacteria ligands eventually lead to NFκB signaling and the expression of inflammatory genes. Control mice received a vaccine consisting of only bacterial TRX in combination with FA. A primer vaccination and three booster vaccinations were given at a two-week interval (Figure 1E). For FA, the complete formulation with the addition of heat-killed and dried Mycobacterium tuberculosis was used for the primer vaccination, while the incomplete adjuvant was used for the booster vaccinations. One week after each vaccination, blood was taken from the tail vein. In week 9, B16F10 melanoma cells were injected subcutaneously into the left flank of the mice, and tumor growth was monitored for up to 15 days. This preclinical mouse study was conducted in duplicate, referred to as Study I and Study II. During both studies, we monitored the bodyweight of all mice to evaluate signs of toxicity (Appendix A). Vaccination was well-tolerated since only minor fluctuations in body weight were detected throughout the experiments.

### 3.2. Montanide Adjuvant Induces a Humoral Immune Response against Extracellular Vimentin

Next, we aimed to characterize the humoral immune response and determined the anti-vimentin antibody titer in all mice after two, three, and four vaccinations (Figure 2A,B). After four vaccinations, FA resulted in the highest antibody titers, reaching statistical significance compared to MN alone (Figure 1A and Appendix A). Importantly, no antibodies against vimentin were seen in mice vaccinated with the control vaccine TRX (Figure 2A). Serum avidity measurements were performed using surface plasmon resonance analysis. This showed comparable avidity of the sera in all three groups (Figure 2C).

The interaction of antibodies and Fc-receptors is crucial for several antibody effector functions such as antibody-dependent cellular phagocytosis (ADCP) and antibody-dependent cellular cytotoxicity (ADCC) [19]. Antibodies of the IgG2b and IgG2c isotype were the dominant subclasses induced after vaccination with either of the three adjuvants used and are known to bind to all four FcγR types. FA and MN-C induced a significantly higher IgG2c titer compared to MN (Figure 2D and Appendix A). While FA and MN-C-vaccinated mice showed that 45–55% of total immunoglobulins is of subclass IgG2c, this is only 10% for MN-vaccinated mice (Appendix A). In contrast, the highest amount of IgM antibodies were found in the MN group, indicating that MN alone might have a lower potential to induce class switching towards IgG (Figure 2D and Appendix A). Antibodies of the IgG3 isotype, representative of a Th1 response [20], were only present in a low amount in all adjuvant groups. In addition, we analyzed the TRXtr-specific antibody titers, and no differences were observed among the three TRXtr-Vim-vaccinated groups (Figure 2E).

The serum level of the pro-inflammatory cytokine IL-6 in the serum of mice vaccinated with FA and MN-C was comparable, while the level in MN-vaccinated mice was significantly lower (Figure 2F). This cytokine is known to be associated with a Th2-mediated humoral immune response. The anti-inflammatory cytokine IL-10 was detectable in 35% of sera of the mice in the FA group, while this percentage was only 16% and 15% for MN and MN-C, respectively. The serum levels of the pro-inflammatory cytokines TNFα or IFNγ, of which the latter is known to play an important role in the cellular immunity involving the activation of cytotoxic T cells, were comparable among the three adjuvant formulations (Figure 2F). To investigate the cellular immunity toward vimentin, we isolated splenocytes from mice vaccinated with either TRX or TRXtr-Vim in combination with Freund’s adjuvant. Restimulation of splenocytes from TRXtr-Vim-vaccinated mice with recombinant endotoxin-free vimentin protein resulted in the production of TNFα and IFNγ in CD4+ T cells, while CD8+ T cells were unaffected (Figure 2G and Appendix A). Enhanced cytokine production of CD4+ T cells after vimentin stimulation was specific for splenocytes isolated from TRXtr-Vim-vaccinated mice (Figure 2H). These data indicate that cellular immunity involving anti-vimentin CD8+ cytotoxic T cells is not induced by vaccination with the TRXtr-Vim conjugate vaccine. Therefore, cellular immunity was not further evaluated for other adjuvant compositions.

In conclusion, the Montanide adjuvant efficiently induced a humoral immune response toward vimentin. Based on the amount and type of antibodies generated, the formulation of Montanide supplemented with CpG results in a highly similar humoral immune response compared to FA.

### 3.3. Vaccination against Vimentin with Montanide Adjuvant Inhibits B16F10 Tumor Growth and Reduces Vessel Density

In two independent experiments, mice vaccinated with FA, MN, or MN-C showed significant tumor growth inhibition, as compared to control vaccinated mice (Figure 3A–C and Appendix AA–D). Quantification of the blood vessels in the tumor tissues resulted in decreased vessel density in all TRXtr-Vim-vaccinated mice (Figure 3D), suggesting that vaccination against extracellular vimentin is indeed successfully targeting the tumor vasculature.

### 3.4. Vaccination against Vimentin Boosts Immune Cell Infiltration into the Tumor

We previously showed that TRXtr-Vim vaccination, when combined with FA, leads to an enhanced immune cell infiltration into the tumor [9]. To investigate whether vaccination with Montanide adjuvant had a similar effect, we quantified the number of immune cells (CD45+) in paraffin-embedded tumor tissues, derived from in vivo Study II. Indeed, vaccination with either FA, MN-C, or MN resulted in a significant increase in the number of immune cells in the tumor. Furthermore, the addition of CpG resulted in a significantly increased infiltration compared to MN alone (Figure 4A,B).

Next, we investigated the infiltration of T cells (CD3+) and myeloid cells (CD11b+). We observed a significant increase in T cell infiltration in all three TRXtr-Vim-vaccinated groups, regardless of the adjuvant used (Figure 4C,D). Regarding the myeloid cell compartment, only MN-vaccinated (*p* < 0.05) and MN-C-vaccinated (*p* = 0.15) mice showed an increased CD11b+ immune cell infiltration in the tumor (Figure 4E,F) compared to control mice.

Finally, we performed linear regression analyses to investigate whether the presence of immune cells was correlated with the vessel density and final tumor volume (Figure 5). Interestingly, the number of CD45+, CD3+, or CD11b+ immune cells were all negatively correlated with the quantified CD31+ vessel density in the tumors (Figure 5A). While no significant correlation could be found between the final tumor volume and the total number of immune cells (CD45+) (Figure 5B, *p* = 0.1995), the number of T cells was negatively correlated with the final tumor volume (Figure 5B, *p* = 0.0054).

In conclusion, especially MN-C-vaccinated mice showed an enhanced immune cell infiltration into tumors. Since the presence of a high number of immune cells correlates with a low vessel density, these immune cells probably contribute to the decreased vessel density in TRXtr-Vim-vaccinated mice.

## 4. Discussion

We have recently shown that extracellular vimentin is overexpressed and secreted by tumor endothelial cells and that anti-vimentin antibodies are capable to inhibit tumor growth in several in vivo murine tumor models [9]. It was demonstrated that vaccination with the iBoost conjugate TRXtr-Vim vaccine generates a potent humoral immune response when coadministered with the gold standard Freund’s adjuvant (FA) (Figure 1C). Although FA is highly immune stimulatory, it is not approved for clinical use due to its toxicity. Therefore, there is a high need to identify a non-toxic alternative that generates a comparable immune response, leading to efficient anti-cancer activity. In the current study, we observed that the formulation of Montanide ISA 720 supplemented with CpG oligodeoxynucleotide 1826, but not Montanide alone, results in a similar humoral immune response towards vimentin, as compared to FA. We detected comparable anti-vimentin antibody titers in the serum of vaccinated mice (Figure 2A) and could confirm that both adjuvants resulted in the generation of IgG2b and IgG2c antibodies. Since these isotypes can bind to all four types of FcγR types, they play a major role in antibody-dependent cellular phagocytosis (ADCP) and antibody-dependent cellular cytotoxicity (ADCC) [19].

To investigate whether the addition of CpG was necessary for a potent anti-vimentin immune response, we included one experimental group in which mice were vaccinated with TRXtr-Vim and Montanide adjuvant (MN) and one group where the adjuvant was supplemented with CpG (MN-C). Unmethylated CpG motifs are abundantly present in prokaryotic DNA, which is released during bacterial infections [21], generating strong immune stimulation. These unmethylated CpG motifs can bind to TLR9 and activate several pathways such as IFN type I signaling [15]. In mice, the TLR9 is expressed by monocytes, macrophages, and conventional dendritic cells [21], while in humans mainly B cells and plasmacytoid dendritic cells express the receptor [22]. We observed that vaccination with MN alone resulted in lower IL-6 production (Figure 2F), lower immune cell infiltration (Figure 4A), and a lower percentage of IgG2c antibodies (Appendix A) compared to MN-C vaccination. Interestingly, IL-6 has been described to have an important role in the differentiation of B cells into antibody-producing plasma cells, a crucial step in our vaccination strategy [23]. Together these data indicated that CpG addition has a beneficial effect on the immune response following TRXtr-Vim vaccination. This effect might partly be explained by an enhanced maturation and activation of the dendritic cells that capture the protein vaccine after injection. Our data are in accordance with a paper by Johansson et al., in which several commonly used adjuvants were compared in their potential to break self-tolerance against immunoglobulin E (IgE) [24]. They showed that vaccination with the well-known adjuvant Alum did not induce antibodies against the self-antigen IgE. However, they did detect anti-IgE antibodies in the serum of rats vaccinated with the biodegradable adjuvant Montanide ISA 720. Furthermore, Huijbers et al. previously showed that Montanide ISA 720, supplemented with CpG, was also successful in the induction of antibodies against ED-B [13,15] and ED-A [8].

In addition to CpG, there are many more immunostimulatory compounds that can be used during vaccination. For example, Freund’s complete adjuvant is complemented with heat-killed and dried Mycobacterium tuberculosis (MT). Mycobacteria ligands can bind a broad range of receptors such as TLR2, TLR4, TLR8, and TLR9 [18,25]. When characterizing the humoral response after vaccination with Montanide supplemented with MT, antibody titers against vimentin were comparable to MN-C-vaccinated mice (data not shown). Since MT was not superior to CpG and no clinical data is yet available on Montanide ISA 720 supplemented with MT, we did not continue with this adjuvant composition in subsequent experiments. Ringvall et al. also directly compared the potency of CpG, double-stranded RNA polyC:poly G (TLR3 ligand), and muramyl dipeptide (NOD2 ligand) [26], three commonly used compounds in vaccine adjuvants. They showed that vaccination with Montanide supplemented with CpG resulted in the highest level of anti-self IgE antibodies, with levels comparable to FA.

The anti-vimentin antibodies that are generated as a result of vaccination with a potent adjuvant are thought to play a major role in the observed reduced vessel density and anti-tumor effect. When anti-vimentin antibodies bind their target in the tumor vasculature, the Fc region of the antibody can bind to an activating FcγR on the surface of several types of immune cells, such as macrophages, neutrophils, and NK cells. The activation of immune cells can trigger effector functions, which eventually leads to tumor vessel destruction, as previously suggested for ED-B [11]. When quantifying the number of blood vessels in B16F10 tumor sections of TRXtr-Vim-vaccinated mice, we indeed found a significantly decreased vessel density (Figure 3D). Interestingly, a decreased vessel density was negatively correlated with an enhanced number of CD45+ immune cells in the tumor (Figure 5A). Increased infiltration of immune cells after vimentin vaccination has recently also been shown by van Beijnum et al. [9] and can potentially be explained by several theories. Firstly, vimentin vaccination upregulates intercellular adhesion molecule (ICAM)-1 expression in the tumor vasculature, which can bind to the leukocyte function-associated antigen (LFA)-1 integrin on leukocytes, facilitating their transmigration into the tumor. Secondly, the P-selectin glycoprotein ligand 1 (PSGL-1) on leukocytes is also involved in binding to the endothelium by binding to its receptor P-selectin. Lam et al. showed that neutrophil adhesion to endothelial cells is reduced in the presence of vimentin, an observation they linked to direct interaction between vimentin and P-selectin [27].

In addition to the total number of immune cells, we also quantified the number of CD3+ T cells in the B16F10 tumors by immunohistochemistry (Figure 4C,D). Vaccination with FA resulted in the strongest T cell infiltration, followed by MN-C. In addition to the upregulation of ICAM-1, it has been shown that TRXtr-Vim vaccination can downregulate vascular PD-L1 expression in B16F10 tumors [9]. Since PD-L1 is involved in T cell exhaustion by interacting with PD-1 on T cells, vimentin vaccination can enhance the fitness of the infiltrating T cells. Interestingly, the number of T cells showed a significant negative correlation with both the vessel density (Figure 5A) and final tumor volume (Figure 5B). Based on CD3 expression, we were unable to differentiate between T helper cells (CD3+CD4+) and cytotoxic T lymphocytes (CD3+CD8+). Flow cytometry analysis on B16F10 tumors after TRXtr-Vim vaccination with Freund’s adjuvant by van Beijnum et al. [9] showed that CD4+ T cells account for approximately 25% of total intratumoral immune cells, while 15% consists of CD8+ cytotoxic T cells. This ratio between CD4+ and CD8+ T cells is similar in tumors after TRXtr-Vim and TRX vaccination. However, the general increase in CD3+ T cells after TRXtr-Vim vaccination (Figure 4C), leads to the hypothesis that both subsets are enriched in the tumor microenvironment after vimentin vaccination. T helper cells can, in addition to their important role in activating vimentin-specific plasma cells, secrete pro-inflammatory cytokines such as IFNγ and TNFα [28]. *Ex vivo* stimulation of splenocytes from TRXtr-Vim-vaccinated mice with recombinant vimentin protein showed that no vimentin-specific cytotoxic T cells were induced by vimentin vaccination (Figure 2G,H). However, cytotoxic T cells can still play a major role by attacking tumor cells directly. The enhanced infiltration of T cells after TRXtr-Vim vaccination shows potential for several types of combination therapies, such as checkpoint inhibition or Chimeric Antigen Receptor (CAR)-T cells [29].

The quantification of the number of CD11b+ immune cells showed that the highest infiltration of myeloid cells was observed in mice vaccinated with MN-C and MN (Figure 4E,F). This CD11b+ immune cell population can contain a large number of different myeloid immune cell subsets, such as monocytes, macrophages, neutrophils, and myeloid-derived suppressor cells (MDSCs). Also NK cells can express CD11b on their cell surface [30]. Based on previous experiments performed by van Beijnum et al., we know that TRXtr-Vim vaccination can enhance the infiltration of several anti-tumor immune subsets such as DCs and NK cells. Importantly, we also observed a reduction in the immunosuppressive MDSCs in the tumors from TRXtr-Vim-vaccinated mice [9]. Since these MDSCs are known to hinder the anti-tumor effects of NK cells, B cells, and T cells, a reduction in this immune subset shows that TRXtr-Vim vaccination generates a less immunosuppressive tumor microenvironment [31]. However, further studies are needed to pinpoint which of these subsets is most abundantly present in the tumor tissues after vaccination with Montanide adjuvant.

Finally, it is important to mention that the Montanide ISA 720 adjuvant has already been tested in several clinical trials. Vaccination against a malaria antigen was safe and well-tolerated, with only mild local reactions at the injection site [32]. Montanide ISA 720 has also already been investigated in several cancer vaccines, such as peptide-based vaccines for melanoma with [33] or without the addition of human CpG 7909 [34]. In these melanoma patients, redness at the injection sites was observed. In addition, two patients showed extensive redness and swelling over their lower abdomen, which resolved within two weeks after immunization [34]. In the current murine studies, we did not observe signs of toxicity associated with the Montanide adjuvant, with or without the addition of CpG, indicating that the adjuvant is also well-tolerated in combination with the TRXtr-Vim vaccine. In addition, we previously extensively studied the potential toxicity of vimentin vaccination in both mice and dogs [9]. We did not observe significant weight loss in mice after TRXtr-Vim vaccination, even when kept hyperimmune for up to a year. Secondly, the TRXtr-Vim vaccine had no effect on wound healing. Finally, the vaccine is well-tolerated in a clinical study in client-owned dogs with spontaneous bladder cancer. We did not observe systemic adverse events, but there were local mild reactions at the injection site [9].

## 5. Conclusions

In the current study, we aimed to investigate the potential of the Montanide ISA 720 adjuvant in combination with the iBoost-based conjugate vaccine TRXtr-Vim. Compared to the very potent but toxic FA, MN-C vaccination resulted in a humoral immune response, similar to the one induced by FA, with a high abundance of the IgG2b and IgG2c isotypes. In addition, both FA and MN-C vaccination resulted in an enhanced immune cell infiltration (CD45+) and tumor growth inhibition in the B16F10 tumor model. Therefore, we conclude that Montanide ISA 720 supplemented with CpG can replace the toxic Freund’s adjuvant in a conjugate anti-cancer vaccine. Since the adjuvant Montanide ISA 720 alone and in combination with the human CpG 7909 sequence is already being used in several clinical trials, the results of this study facilitate further evaluation of the TRXtr-Vim vaccine in the clinic.

## Figures and Tables

**Figure 1 cancers-14-02593-f001:**
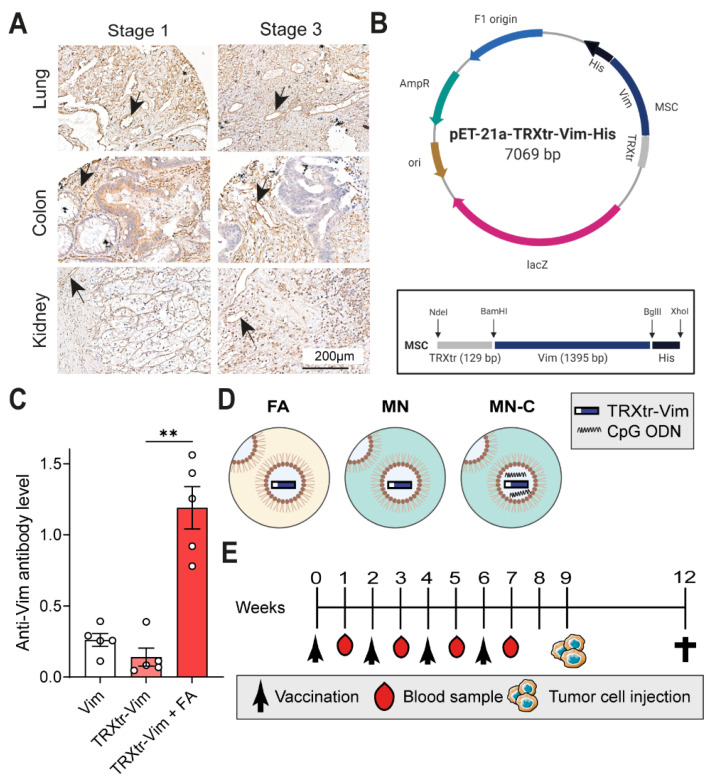
Vimentin expression in cancer tissues and study setup. (**A**) Representative images of vimentin expression in human tumor tissues from in-house developed TMA, at a 100x magnification. Arrows indicate a vimentin-positive blood vessel present in the tissue. (**B**) The pET21a expression vector for production of TRXtr-Vim. The DNA sequences of the different protein constructs (insert) were inserted between the restriction sites Nde1, BamH1, and Xho1 in the multiple cloning site, as indicated in the black rectangular. (**C**) Anti-vimentin total immunoglobulin (Ig) levels in serum after the third vaccination measured at a 1:100 serum dilution by ELISA at OD655 nm. Vaccination was performed with either only vimentin protein (Vim), TRXtr-Vim protein (TRXtr-Vim), or TRXtr-Vim combined with Freund’s adjuvant (TRXtr-Vim + FA). (**D**) Schematic overview of the treatment groups. FA is Freund’s adjuvant, MN is Montanide ISA 720 adjuvant, and MN-C is Montanide ISA 720 supplemented with CpG, a TLR9 agonist. (**E**) Schematic overview of the vaccination and tumor growth schedule. Mice were vaccinated four times at two-week intervals. One week after each vaccination, a blood sample was taken. In week 9, B16F10 melanoma cells were injected subcutaneously and tumor growth was monitored. Figure created with BioRender.com. ** *p* < 0.01.

**Figure 2 cancers-14-02593-f002:**
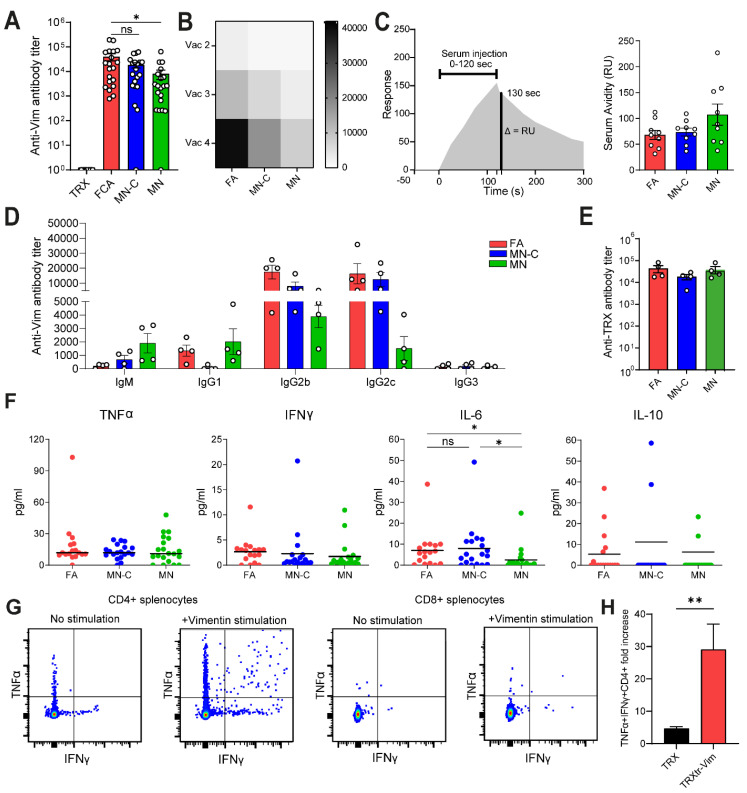
Characterization of the humoral immune response after vaccination with different adjuvant compositions. (**A**) Anti-vimentin total immunoglobulin (Ig) titers for each mouse in serum after the fourth vaccination using ELISA. Antibody titers were calculated based on the intersection between the titration curve and the threshold of OD655 of 0.2. Data from Study I and Study II wee pooled. (**B**) Anti-vimentin total immunoglobulin (Ig) titers for each vaccine group in serum after second, third, and fourth vaccination using ELISA. Values present the mean titer within each experimental group (*n* = 20). (**C**) Antibody avidity determination of anti-vimentin antibodies in serum using Biacore. Sera after the fourth vaccination were taken from Study I for analysis, and the amount of serum/antibodies input was normalized based on ELISA titers. Serum Avidity (RU) was determined as the response 10 s after serum injection. (**D**) Analysis of vimentin-specific IgG subclasses and IgM after four vaccinations using ELISA. Includes data of both Study I and Study II, with sera of 5 mice pooled per dot. (**E**) Anti-TRX total immunoglobulin (Ig) titers for each vaccine group from Studies I and II in serum after the fourth vaccination using ELISA, with sera of 5 mice pooled per dot. (**F**) Cytokine profiling in serum after fourth vaccination using LegendPlex. (**G**) Representative dot plots depicting IFNγ and TNFα expression in CD4+ T cells and CD8+ T cells isolated from spleens of TRXtr-Vim-vaccinated mice in combination with FA. Cells were either unstimulated or stimulated with recombinant vimentin protein for 48 h. Brefeldin A was added 6 h before cell harvesting for FACS analysis. (**H**) The fold increase in TNFα+IFNγ+ CD4+ T cells after vimentin stimulation as compared to cells incubated without vimentin protein. CD4+ T cells were either isolated from spleens of TRX-vaccinated mice (*n* = 5) or TRXtr-Vim-vaccinated mice (*n* = 10). * *p* < 0.05, ** *p* < 0.01.

**Figure 3 cancers-14-02593-f003:**
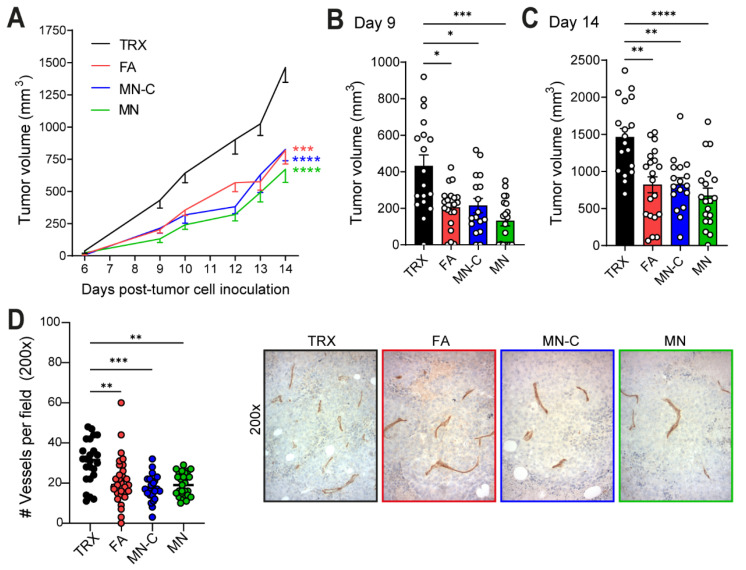
B16F10 melanoma tumor growth and tumor vessel density after vaccination against extracellular vimentin. (**A**) Pooled tumor growth curves of B16F10 melanoma in vaccinated mice of Studies I and II. Tumor growth curves are presented as mean + SEM and were compared by two-way ANOVA. (**B**) Tumor volume 9 days after tumor-cell inoculation of both studies combined. Bars represent mean + SEM. (**C**) Tumor volume 14 days after tumor cell inoculation of both studies combined. Bars represent mean + SEM. (**D**) Vessel density in B16F10 tumors derived from Study II as determined by CD31+ vessels by IHC. Each dot represents one microscopic field. Representative images of CD31 staining for each group shown on the right. * *p* < 0.05, ** *p* < 0.01, *** *p* < 0.001, and **** *p* < 0.0001.

**Figure 4 cancers-14-02593-f004:**
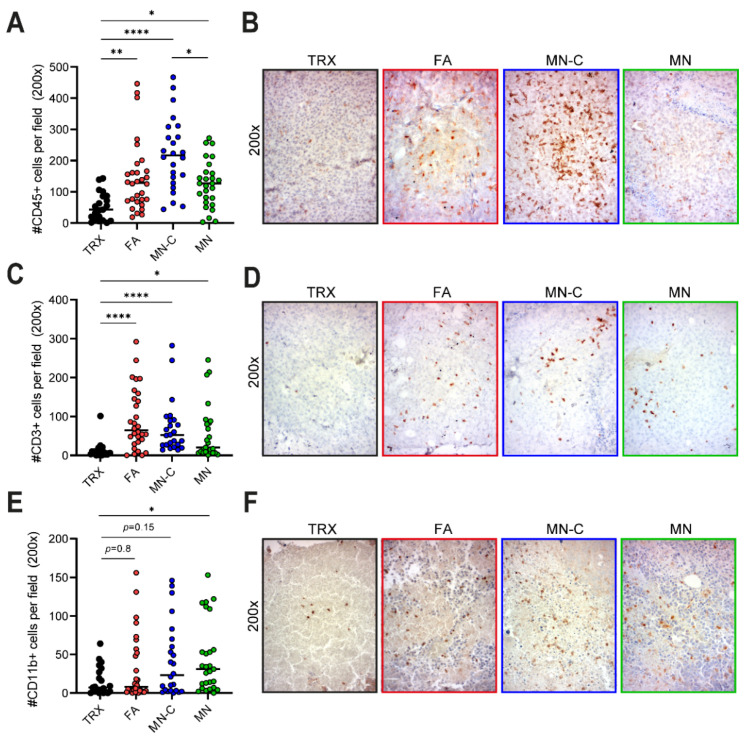
Immune cell infiltration in B16F10 melanoma after vaccination. (**A**) CD45+ immune cells present in B16F10 tumors derived from Study II as determined by IHC. Each dot represents one microscopic field. (**B**) Representative images of CD45 staining for each group at 200× magnification. (**C**) CD3+ immune cells present in B16F10 tumors derived from Study II as determined by IHC. Each dot represents one microscopic field. (**D**) Representative images of CD3 staining for each group at 200× magnification. (**E**) CD11b+ immune cells present in B16F10 tumors derived from Study II as determined by IHC. Each dot represents one microscopic field. (**F**) Representative images of CD11b staining for each group at 200× magnification. * *p* < 0.05, ** *p* < 0.01, and **** *p* < 0.0001.

**Figure 5 cancers-14-02593-f005:**
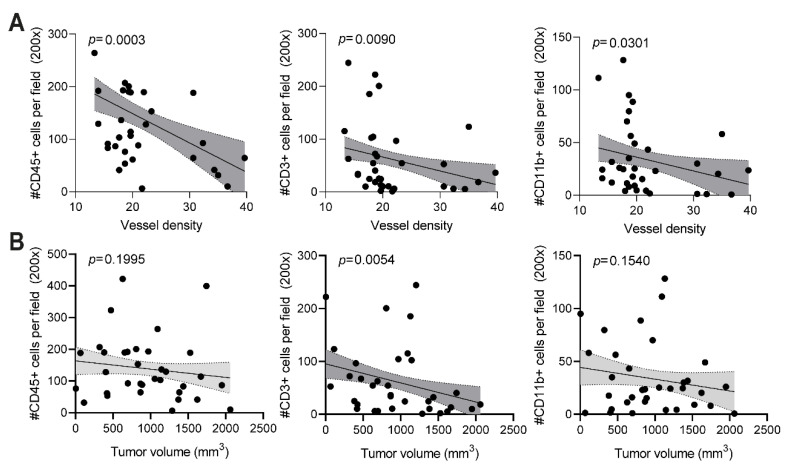
Enhanced immune cell infiltration correlates with vessel density and tumor growth (**A**) Significant correlation between the number of CD45+, CD3+, and CD11b+ immune cells and vessel density (number of CD31+ vessel per microscopic field). (**B**) Correlation between CD45+, CD3+, and CD11b+ immune cell infiltration and final tumor volume at day 14 (mm^3^). Significant correlations (*p* < 0.05) are indicated with the dark grey area.

## Data Availability

The data presented in this study are available on request from the corresponding author.

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
