# Peer review of "Cancer Vaccination against Extracellular Vimentin Efficiently Adjuvanted with Montanide ISA 720/CpG"

_cancers, 2022, doi:10.3390/cancers14112593_

Round 1

Reviewer 1 Report

This is a well crafted study by van Loon et al, evaluating the performance of an anti-vimentin tumor vaccine with a novel form of squalene adjuvant Montanide ISA 720/CpG. The topic is of high interest and relevant. The Reviewer suggests the following improvements for the manuscript before publication:

Minor Remark.

  • Please improve the quality of IHC images (or enchance the pdf's quality)

Major Remarks:

  • The authors assessed CD3+ pan-T-cell marker in tumors of vaccinated experimental animals. However the most potent and general anti-tumor action is attributed to CD8+ cytotoxic T-cells. What about the ratio of this T-cell subgroup? The Authors state: "Due to the strict deletion of autoreactive T cells in the thymus during T cell development, it is unlikely that vimentin-specific cytotoxic T lymphocytes are directly involved in the disruption of the tumor vasculature" --> true, but adjuvant effects and the diminishment of the potentially immunosuppresive role of vimentin in the tumor vasculature can attract more CD8+ T-cells (not anti-vimentin) to attack tumor cells. Showing this would improve the results a lot.
  • Is there any change in experimental tumors regarding immunosuppressive TME cells, like tumor-associated macrophages (CD68+) or MDSCs (CD33+) after anti-vimentin vaccination with other adjuvants? Please elaborate more in the discussion.
  • The Authors described in details the potential AEs of MN adjuvants described by recent studies, but they do not discuss AEs attributed to anit-vimentin immunisation. Is there absolutely no long-term adverse events observed due to anti-vimentin vaccination? It is difficult to believe that immunization against a so general own ECM molecule has no long-term side effect at all. Please elaborate in the discussion.
  • Assessment of PD-L1/PD1 expression in a melanoma model study is essential. Most future cancer vaccine therapies will be administered together with currently used ICI-therapy, therefore the Authors cannot leave out this topic. Assessment of PD-L1/PD1 expression before/after immunization need to be assessed (regardless it yields any new results or not), and a paragraph interpreting the topic in the discussion is also needed.  

Reviewer 2 Report

van Loon et al tested usage of a vimentin vaccine the group has previously developed and reported in combination with a clinically feasible vaccine adjuvant, Montanide ISA 720 with and without combination with CpG. Overall, the quality of the work is good and the main conclusions are supported by the results. However, aside from the importance of this work to the clinical translation of this approach, the significance of the work is limited.

  1. The authors should consider briefly presenting the clinical feasibility considerations of montanide ISA in the introduction or abstract. It was not clearly presented that the adjuvant approach would be safer than Fruend’s adjuvant until the last paragraph of the discussion. While weights were recorded from mice after various compositions of vaccine, there no convincing data showing that these adjuvants are more favorable than Fruend’s adjuvant from a toxicity standpoint.

  1. The authors have another paper currently in revision at Nature communications that this work references (#9). Since this work is not publicly available, it is not possible to determine if the work here extends beyond what is shown in that publication.

  1. This study is relatively limited from a mechanistic standpoint. For example, B cell knockout mice could be used to test if the vaccine functions by generating antibodies against vimentin vs T cell mediated effects. Passive transfer of serum could be used to determine if antibodies mediate the observed effects. Are the effects dependent on FCgamma receptors (FCgamma Receptor KO mice are available)?

  1. It is unclear how this approach would be superior to passive immunity with an anti-vimentin antibody; or clinically available VEGF blocking approaches like avastin. It is anticipated that from a safety standpoint, passive immunity would be optimal as opposed to breaking immune tolerance to a wt self-antigen. Due to this lacking evidence, it is unclear whether this approach advances the field or represents a clinically feasible approach.

  1. Fig2A- control vaccination antibody titers should have been tested/shown somewhere to show the vaccines do in fact generate new antibody responses as opposed to modulating them.

  1. Fig 2F- serum levels of cytokines cannot be meaningfully interpreted without a control group included. IL6 and other cytokines are present in serum at baseline.

  1. T cell immunity was not addressed directly, and it is not sufficient to say that self-antigen specific CD8 T cells cannot be generated (In discussion). Could be addressed with further mechanistic study, as in point 3

  1. Some hyperbole in the text should be tempered: e.g. “great promise” (abstract)

Round 2

Reviewer 1 Report

The Authors have assessed all the raised concerns and the Reviewer would like to congratulate for this important study. Work should be continued and proceed for Phase I human trials. 

Reviewer 2 Report

The authors have addressed all of my concerns.